# Institutionalization of limited obstetric ultrasound leading to increased antenatal, skilled delivery, and postnatal service utilization in three regions of Ethiopia: A pre-post study

**Hailemariam Segni Abawollo[1‡], Mesele Damte Argaw** [1,2‡*], **Zergu Taffesse Tsegaye[1‡],
Ismael Ali Beshir[1‡], Asfaw Adugna Guteta[1◉], Atrie Fekadu Heyi[1◉],
Birhan Tenaw Mengesha[1◉], Tsega Teferi Mamo[1◉], Zenawork Kassa Gebremedhin[1◉],
Heran Demissie Damte[1◉], Mengistu Asnake Kibret[3◉], Binyam Fekadu Desta[1]**

1 USAID Transform: Primary Health Care Activity, JSI Research & Training Institute Inc., Addis Ababa, Ethiopia, 2 USAID Surveillance for Malaria Elimination Activity, PATH, Addis Ababa, Ethiopia, 3 USAID Transform: Primary Health Care Activity, Pathfinder International, Addis Ababa, Ethiopia

◉ These authors contributed equally to this work.
‡ HSA, MDA, ZTT and IAB also contributed equally to this work.
* mdamte5@gmail.com

## Abstract

### Background

A minimum of one ultrasound scan is recommended for all pregnant women before the 24th week of gestation. In Ethiopia, there is a shortage of skilled manpower to provide these services. Currently, trained mid-level providers are providing the services at the primary healthcare level. The aims of this study were to compare antenatal care 1 (ANC1), antenatal care 4 (ANC4), skilled birth attendance (SBA), and postnatal care (PNC) service utilization before and after institutionalizing Vscan limited obstetric ultrasounds at semi-urban health centers in Ethiopia.

### Methods

A pre and post intervention observational study was conducted to investigate maternal and neonatal health service utilization rates before and after institutionalizing Vscan limited obstetric ultrasound services, between July 2016 and June 2020. The data were extracted from 1st August– 31st December 2020.

### Results

The observed monthly increase on the mean rank of first ANC visits after the introduction of Vscan limited obstetric ultrasound services showed a statistically significant difference at KW-ANOVA H (3) = 17.09, P = 0.001. The mean rank of fourth ANC utilization showed a statistically significant difference at KW- ANOVA H (3) = 16.24, P = 0.001. The observed mean

**Data Availability Statement:** The data used for this paper is attached as supplement file.

**Funding:** The USAID Transform: Primary Health Care project is a technical assistance program to support the Government of Ethiopia. The project is funded by the United States Agency for International Development (USAID), under cooperative agreement number of AID-663-A-17-00002 and is managed by Pathfinder International and JSI Research & Training Institute Inc. in Ethiopia. The project purchased Vscan limited obstetric ultrasound machines which were distributed with seed supplies to 100 health centers. The cost of basic ultrasound training was also covered by the project. In addition, the project covered the cost of data collectors. The funder provided support in the form of salaries for authors HSA, MDA, ZTT, IAB, AAG, AFH, BTM, TTM, ZKG, HDD, MAK, and BFD. This technical report is made possible by the generous support of the American people through USAID. However, the funding body had no role in the study design, data collection, analysis, decision to publish, or preparation of the manuscript.

**Competing interests:** The authors have declared that no competing interests exist.

rank in skilled birth attendance (SBA) showed a statistically significant positive difference using KW-ANOVA H (3) = 23.6, P<0.001. The mean rank of increased utilization in postnatal care showed a statistically significant difference using KW-ANOVA H (3) = 17.79, P<0.001.

## Conclusion

The introduction of limited obstetric ultrasound services by trained mid-level providers at the primary healthcare level was found to have improved the utilization of ANC, SBA, and post-natal care (PNC) services. It is recommended that the institutionalization of limited obstetric ultrasound services be scaled up and a further comparative study between facilities with and without ultrasound services be conducted to confirm causality and assess effects on maternal and perinatal outcomes.

## Introduction

Globally, in the year 2017, about 295,000 women died during and following pregnancy and childbirth. Most of these deaths (94%), occurred in low-resource settings; about two-thirds (196,000) of maternal deaths occurred in the sub-Saharan Africa region (SSA) [1]. In Addition, the World Health Organization (WHO) estimated that 2.5 million children died within the first month of their life in the year 2018 [2]. Furthermore, 2.6 million babies were stillborn. The majority of the deaths (99%), occurred in low and lower-middle-income countries, with half of these deaths happening at home [3].

The United Nations (UN) member countries have pledged to reduce maternal mortality ratios by at least two thirds by the year 2030 from their 2010 baseline. Even sub-Saharan countries, with heavier burdens, are expected to achieve greater reductions. Similarly, for the same period, countries are aiming to reduce neonatal mortality ratios to as low as 12 per l,000 live births [4]. The stated ambitious commitments of states were supplemented with proven maternal and neonatal health interventions on services. These services are offered to women during pregnancy, childbirth, and postpartum periods, whereas a healthy neonatal period starts from 22 completed weeks of gestation and continues to the first month of life [5].

The World Health Organization (WHO) works towards a world where every pregnant woman and newborn receives quality care throughout the pregnancy, childbirth, and postnatal periods [6]. Despite the increasing efforts to address challenges, around 40 percent of fetal, neonatal, and maternal deaths occur during the intrapartum period or on the day of birth [7]. Early identification, confirmation, and arrangement of referral services for high-risk health conditions reduce morbidities and mortalities in low and middle-income countries [8, 9]. These conditions all rely on offering at least one ultrasound scanning service for every pregnant woman before the 24th week of gestation for confirming the diagnoses [6]. However, access to ultrasound services is limited in low-resource settings due to human resource constraints and lack of availability of the required technology [10]. Optimizing the capacity of existing human resources and distribution of tasks and responsibilities among various health professionals will improve access to, and quality and equity of health services [11].

During the year 2020, there were 95 radiologists and 462 obstetricians/gynecologists available to provide obstetric ultrasound services in Ethiopia, which has a population of over 110 million. In addition, most antenatal care (ANC), skilled birth attendance (SBA) i.e.,

institutional delivery, and postnatal care (PNC) services are carried out in rural health centers where ultrasound services are not available [12].

Studies have documented improvements in maternal and neonatal health service utilization and favorable pregnancy outcomes for both mother and baby as well as improvements in quality of services in remote places around the world, as a result of availing obstetric point of care ultrasound services [13–26]. The United States Agency for International Development (USAID) Transform: Primary Health Care project institutionalized antenatal limited obstetric ultrasound services in 100 health centers in Ethiopia. In addition, the Ethiopian Ministry of Health (MoH) has developed a plan to scale up obstetric ultrasound services in 1,000 health centers by 2025 [27]. Despite this, ANC1, ANC4, SBA, and PNC service utilization based on institutionalizing limited obstetric ultrasounds has not been assessed. Therefore, this study aims to compare ANC1, ANC4, SBA, and PNC service utilization before and after institutionalizing Vscan limited obstetric ultrasounds at semi-urban health centers in Ethiopia. To the authors' knowledge, this study is the first of its kind in the country and will be used to guide policy on the area.

## Operational definitions

### Institutionalization of limited obstetric ultrasound services

The USAID Transform: Primary Health Care project implemented a step-by-step process of institutionalizing limited obstetric ultrasound services at the health center level. The processes followed consisted of laying down the legal framework, capacity building service initiation, and transitioning to the public health sector. A detailed description of the four implemented steps is presented below.

- The first step was working on the legal framework and developing governing documents as a foundation of institutionalizing ultrasound services. Therefore, the project organized a series of preliminary discussions with policy makers, program managers, and healthcare providers. The output of these discussions was identification of the roles and responsibilities of stakeholders, development of course syllabus, and identification of the 100 health centers and mapping out of their referral networks.

- The second step was dedicated to capacity building of mid-level health professionals i.e., BSc. nurses and midwives on operating limited obstetric ultrasound scanning, diagnosis, and management services, orientation of community health workers on promoting health literacy of pregnant women, and preparation of service sites with rooms, supplies, and equipment.

- The third step was the initiation of fee exempted antenatal ultrasound services in the 100 selected health centers. In addition, the project maintained and ensured the quality of services through mentoring, coaching, and conducting supervisions.

- The fourth step was enabling a full transition to the public health system. The project trained at least two mid-level health professionals per health center and conducted implementation research on barriers and facilitators of ultrasound services. Using the results, the project advocated for continuous refilling of supplies, training of more providers, and scaling up at new sites using public sector resources.

## Materials and methods

### Design, duration, and setting

A pre- and post-intervention observational study was conducted to understand maternal and neonatal health service utilization rates before and after institutionalizing Vscan limited

obstetric ultrasound services between July 2016 and June 2020. The actual data extraction was made from 1st August to 31st December 2020. The study was conducted in Amhara, Oromia, and Southern Nations, Nationalities and Peoples' (SNNP) regions. These regions were selected due to the presence of Vscan limited obstetric ultrasound services in the selected health centers, for over two years. A health center is a public health facility within the primary healthcare system of Ethiopia, serving up to 25,000 people and has a mandate to provide promotive, preventive, curative, and rehabilitative outpatient care including basic laboratory and pharmacy services with a capacity for 10 beds for emergency and delivery services [12].

### Intervention

USAID funded the USAID Transform: Primary Health Care project (June 2016-June 2020) to support the MoH in line with its long-term goal of preventing child and maternal deaths (PCMD) [28]. To improve access to, and quality and equity of basic maternal and neonatal health services, the project, in collaboration with its technology partner General Electric Healthcare introduced Vscan access—a small portable ultrasound device—for obstetric scanning by trained mid-level healthcare providers [29]. The project strategizes to increase SBA through introducing ANC limited obstetric ultrasound services and improving the proper management of identified complications of pregnancies in referral health facilities, all of which contribute to maternal, fetal, and neonatal positive health outcomes [28, 30].

Vscan ultrasound machines with seed supplies were provided to each of the targeted 100 health centers after successfully providing basic limited ultrasound classroom and practical hands-on training for 10 days. The trainees were 219 mid-level healthcare providers who lacked knowledge and skills on the technology [31]. In addition, the training was supplemented with three sessions of monthly coaching, each lasting for two days, and the provision of virtual real-time feedback. Service initiation was supported through awareness creation and dissemination of information on ultrasound service availability at the health center level using all community engagement platforms including pregnant women conferences, women development army meetings, and house to house visits by health extension workers [28]. To ensure the quality of ultrasound services, a continuous mentorship was carried out by government and project staff. Pregnant women with detected abnormalities during scanning were referred to nearby hospitals for confirmation of diagnoses and further care. The number of women that received ultrasound scanning services during the first, second, third, or more trimester periods within the two years after the introduction of the services were 10,186, 2,974, and 1,509, respectively.

### Study population

Based on the 2007 national census, the projected population of the residents of targeted areas were 1.10 million in the year 2017. At the endpoint (2020), there were about 1.19 million people living within the study areas. Of these, the estimated number of pregnant women eligible for maternal and neonatal health services for the year 2020 were 40,506 (3.4%). The majority (13/30) of the health centers were enrolled from within the Oromia Region. On average, each health center is located 58.6 kilometers away from a referral receiving hospital (**Table 1**).

### Sample size and sampling

The sample size was determined using the rule of thumb recommendation of The Aga Khan Foundation (1997) [32]. Thirty health centers were sampled i.e., 30% of the 100 health centers. The three regions were selected based on the accessibility and functionality of the ultrasound

**Table 1. Characteristics of the study area and population, USAID Transform: Primary Health Care project intervention sites, Ethiopia, July 2016- June 2020.**

| Characteristics | Amhara | Oromia | SNNP | Total |
|---|---|---|---|---|
| Number of health centers | 9 | 13 | 8 | 30 |
| Average distance from a referral receiving facility in kilometers | 59.5 | 70.4 | 38.4 | 58.6 |
| Population | | | | |
| 2017 | 329,038 | 529,008 | 242,815 | 1,100,861 |
| 2018 | 337,822 | 543,129 | 249,298 | 1,130,249 |
| 2019 | 346,841 | 557,628 | 255,953 | 1,160,422 |
| 2020 | 356,100 | 572,513 | 262,785 | 1,191,398 |
| Eligible women | | | | |
| 2017 | 11,873 | 19,270 | 8,858 | 40,001 |
| 2018 | 11,949 | 19,333 | 8,884 | 40,166 |
| 2019 | 12,028 | 19,398 | 8,909 | 40,335 |
| 2020 | 12,106 | 19,464 | 8,936 | 40,506 |

services during the time of data collection. Finally, a simple random sampling technique was applied to identify individual facilities.

## Data collection

Three supervisors and 30 data collectors who are health science professionals were recruited from the targeted three regions. A two-day training on the objectives of the study, data collection techniques, ethical principles, and field pretesting was carried out. The data were extracted from a routine health information management system (HMIS) database using a pre-tested tool. To ensure the quality of data, three trained supervisors with master's degrees in public health were deployed in the field and provided close technical support, with feedback given daily to the data collectors. The supervisors were responsible for checking and rechecking the collected data for completeness and consistency.

The **dependent variables** were summary aggregated continuous data [33] of ANC1, ANC4, SBA, and PNC service reports.

**The independent variables** were years of service.

## Data analysis

The data were entered and cleaned using Microsoft Excel 2016 and exported to SPSS V25 for descriptive and inferential analysis. The service utilization coverages were compared based on institutionalization of Vscan limited obstetric ultrasounds using 1,440 aggregate data collected from the 30 health centers (**S1 File**). In addition, for this study, the steps and procedures of Ross et al., (2013) were adopted [14]. The results of the statistical tests were presented using tables and graphs. To analyze F tests (one-way analyses of variance, ANOVA), the data violated the assumption of homogeneity of variances of a parametric test using Shapiro-Wilk test of normality $P < 0.05$ (**S2 File**). Hence, the Kruskal-Wallis H test or a 'one-way ANOVA on ranks' which is an equivalent non-parametric test was employed to determine statistically significant differences between four groups of independent variables i.e., years of service. The statistical differences were claimed at $P < 0.05$. However, the investigators maintained and ensured the following assumptions of the Kruskal-Wallis H test: (1) the dependent variables (ANC, SBA, and PNC) are interval data, (2) the independent variable has four categories, (3) there was no relationship between observations in each group, and (4) the distribution of scores in each group were not identical [28]. Finally, a statistical analysis was employed to

compare data from twenty-four months (2017 and 2018) preceding the introduction of Vscan limited obstetric ultrasound services and twenty-four months (2019 and 2020) following the intervention. The mean rank monthly first ANC, fourth ANC, SBA, and PNC were compared before and after ultrasound services were initiated using a nonparametric test called the Kruskal-Wallis H test. Post hoc analysis using the KW-ANOVA H mean rank test was conducted with a Bonferroni correction applied. A statistical test result with a *P*-value of <0.0125 indicated the presence of a significant difference between service coverages over four years on ANC1, ANC4, SBA, and PNC.

## Ethical considerations

This study protocol was carried out in accordance with the Declaration of Helsinki [34] and ethical clearance was granted from the JSI Institutional Review Board (IRB), through reference number IRB#20-26E. The IRB determined that this research activity is exempted from human subject oversight. Informed individual written consent was obtained from each health center manager to extract prenatal care information from the routine health information system database. All pregnant women's personal and medical record information were de-identified using ANC record numbers. Informed written consent was obtained from all participants. In addition, personal identifiers of pregnant women were not captured. The limited obstetric ultrasound scanning program was designed without fetal gender determination for ethical reasons and was checked regularly by mentors. The investigators maintained national and international ethical principles including ensuring the anonymity and confidentiality of research participants and collected data throughout the research process.

## Results

The results of this retrospective study are presented as descriptive information on maternal and neonatal health service utilization and difference in the means of ranked ANC1, ANC4, SBA, and PNC data among four groups.

### Maternal and neonatal health service utilization

The average first ANC service utilization rates were 43.6%, 52.8%, 73.7%, and 72.9% for the years 2017, 2018, 2019, and 2020, respectively (Fig 1). In addition, the mean numbers of the first ANC visits were 765 (±SD) and 1126.9 (±SD), for equal reporting periods of pre- and post-ultrasound introductions in the 30 selected health centers (**Table 2**). The average fourth ANC visit coverages were 43.8% and 68.3%, for the years 2017 and 2020, respectively. Similarly, the mean numbers of fourth ANC visits were 534 (±SD) and 777.5 (±SD), during the pre- and post-ultrasound introduction periods, respectively.

### Mean rank comparison of maternal and neonatal health services

A Kruskal-Wallis mean rank test was conducted to determine if there were differences in first ANC, fourth ANC, SBA, and PNC service scores between groups that differed in service years.

The results of a retrospective assessment of maternal and neonatal health service beneficiaries and coverages using four categories in the 30 health centers are presented below (**Table 3**). The mean rank of first ANC visits was 43.62, 52.78, 73.67, and 72.93 for the years 2017, 2018, 2019, and 2020, respectively. The introduction of Vscan limited obstetric ultrasound service shows a statistically significant difference on first ANC visits over four years at KW-ANOVA H (3) = 17.09, P = 0.001. Similarly, the mean rank of fourth ANC was 43.82, 53.18, 76.68, and

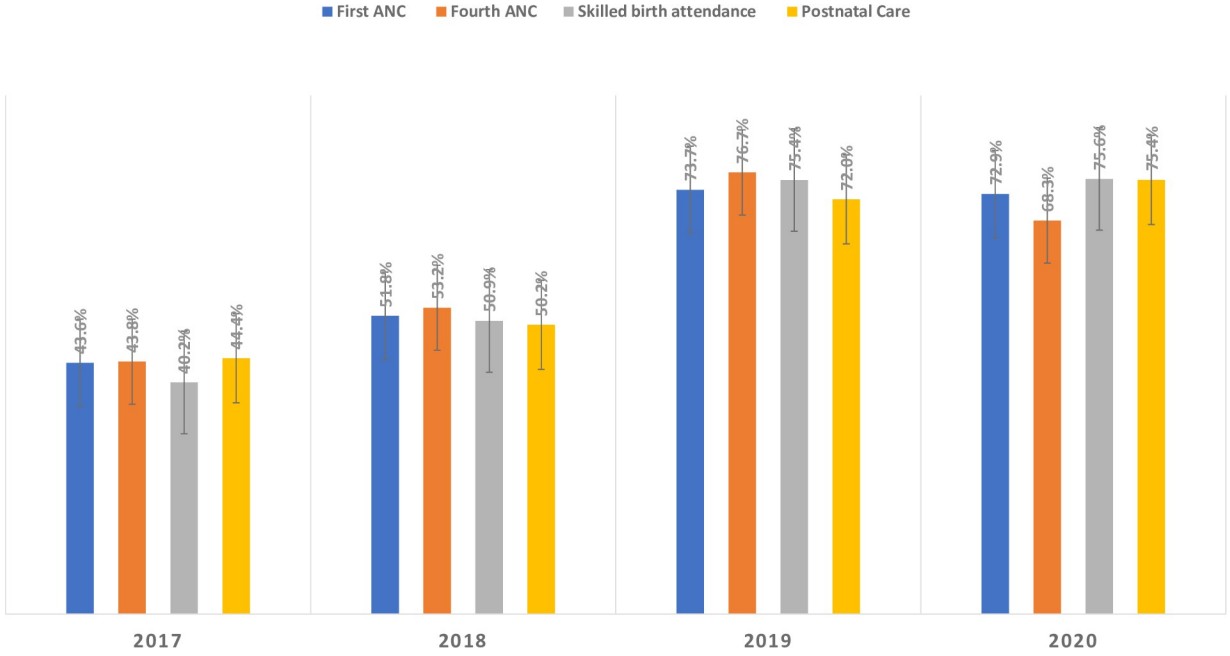

**Fig 1. Bar chart showing first ANC, fourth ANC, SBA, and PNC service utilization coverages prior to, and post the introduction of ultrasound services at health centers, Ethiopia, July 2016-June 2020.**

**Table 2. Number of monthly ANC, SBA, and PNC service utilization in 30 health centers, pre- and post-ultrasound introduction in Ethiopia, July 2016-June 2020.**

| Measurements | Time | Facility- months | Mean (SD) | Median | Min-Max | 95%CI |
|---|---|---|---|---|---|---|
| **Number of first ANC visits** | 2017 | 720 | 705.7 (378.8) | 636.5 | 136–1859 | 564.3–847.2 |
| | 2018 | 720 | 824.3 (512.2) | 625.0 | 251–2253 | 633.1–1015.6 |
| | Pre-ultrasound introduction | 1440 | 765 (450.6) | 625.0 | 136–2253 | 648.6–881.4 |
| | 2019 | 720 | 1130.3 (617.6) | 905.5 | 254–2916 | 899.6–1360.9 |
| | 2020 | 720 | 1123.6 (625.7) | 905.5 | 254–2916 | 889.9–1357.3 |
| | Post-ultrasound introduction | 1440 | 1126.9 (616.5) | 905.5 | 254–2916 | 967.7–1286.2 |
| **Number of fourth ANC visits** | 2017 | 720 | 475.7 (243.8) | 454.5 | 126–964 | 384.7–566.8 |
| | 2018 | 720 | 592.3 (409.4) | 455.5 | 218–1955 | 439.5–745.2 |
| | Pre-ultrasound introduction | 1440 | 534 (339.2) | 454.5 | 126–1955 | 446.4–621.6 |
| | 2019 | 720 | 802.9 (378.5) | 754.0 | 177–2007 | 661.6–944.3 |
| | 2020 | 720 | 752.1 (431.9) | 665.5 | 148–1989 | 590.8–913.4 |
| | Post-ultrasound introduction | 1440 | 777.5 (403.4) | 715.0 | 148–2007 | 673.3–881.7 |
| **Number of SBA** | 2017 | 720 | 423.7 (224.0) | 377.5 | 116–964 | 340.0–507.4 |
| | 2018 | 720 | 552.0 (384.8) | 437.0 | 218–1756 | 408.3–695.7 |
| | Pre-ultrasound introduction | 1440 | 487.9 (318.8) | 393.5 | 116–1756 | 405.5–570.2 |
| | 2019 | 720 | 747.6 (320.8) | 679.5 | 232–1810 | 627.8–867.4 |
| | 2020 | 720 | 775.4 (455.8) | 677.5 | 252–2427 | 605.2–945.7 |
| | Post-ultrasound introduction | 1440 | 761.5 (391.0) | 679.5 | 232–2427 | 660.4–862.5 |
| **Number of PNC visits** | 2017 | 720 | 464.3 (248.9) | 434.0 | 99–983 | 371.3–557.3 |
| | 2018 | 720 | 559.4 (410.4) | 391.5 | 123–1756 | 406.2–712.7 |
| | Pre-ultrasound introduction | 1440 | 511.8 (339.9) | 398.0 | 99–1756 | 424.0–599.6 |
| | 2019 | 720 | 774.6 (454.6) | 652.0 | 201–2339 | 604.8–944.3 |
| | 2020 | 720 | 805.7 (455.9) | 705.5 | 223–2427 | 635.5–975.9 |
| | Post-ultrasound introduction | 1440 | 790.1 (451.6) | 675.0 | 201–2427 | 673.5–906.8 |

**Table 3. Mean rank scores of ANC1, ANC4, SBA, and PNC services and difference in distribution over four years (2017–2020).**

| Services | samples | N = 120 | Mean Rank | Kruskal-Wallis H | df | Asymp. Sig. |
|---|---|---|---|---|---|---|
| ANC1 | 2017 | 30 | 43.62 | 17.09 | 3 | 0.001 |
| | 2018 | 30 | 51.78 | | | |
| | 2019 | 30 | 73.67 | | | |
| | 2020 | 30 | 72.93 | | | |
| ANC4 | 2017 | 30 | 43.82 | 16.24 | 3 | 0.001 |
| | 2018 | 30 | 53.18 | | | |
| | 2019 | 30 | 76.68 | | | |
| | 2020 | 30 | 68.32 | | | |
| Skilled delivery | 2017 | 30 | 40.20 | 23.59 | 3 | 0.001 |
| | 2018 | 30 | 50.90 | | | |
| | 2019 | 30 | 75.35 | | | |
| | 2020 | 30 | 75.55 | | | |
| PNC | 2017 | 30 | 44.42 | 17.79 | 3 | 0.001 |
| | 2018 | 30 | 50.22 | | | |
| | 2019 | 30 | 72.00 | | | |
| | 2020 | 30 | 75.37 | | | |

68.32 in 2017, 2018, 2019 and 2020, respectively. Fourth ANC service utilization shows a statistically significant difference over four years at KW- ANOVA H (3) = 16.24, P = 0.001.

The mean rank SBA scores were 40.20, 50.90, 75.35, and 75.55in the year 2017, 2018, 2019 and 2020, respectively. The observed increased mean rank SBA scores show a statistically significant positive difference using KW-ANOVA H (3) = 23.6, P<0.001. While the mean rank of PNC service scores were 44.42, 50.22, 72.00, and 75.37 in the years 2017, 2018, 2019 and 2020, respectively. The mean rank of PNC service utilization scores show a statistically significant difference using KW-ANOVA H (3) = 17.79, P<0.001.

## Difference in maternal and neonatal health service coverages

The results of pairwise mean rank comparisons with post hoc analysis of the first ANC, fourth ANC, SBA, and PNC services are depicted in the table presented below (Table 4). There were statistically significant positive differences between mean rank scores of ANC1 in samples of the third and first (P = .007) and fourth and first years (P = .005). The mean rank scores of ANC4 have shown statistically positive difference between samples of the third and first years (P = .002). Similarly, the mean rank scores of SBA services have shown a statistically positive difference between samples of the fourth and first (P = .001) and third and first years (P = .001). Furthermore, the mean rank scores of PNC services have shown a statistically positive difference between samples of the third and first years (P = .001).

## Discussion

This study has shown that the institutionalization of limited obstetric ultrasound services by trained mid-level providers at semi urban health centers significantly improved the utilization of prenatal, intrapartum, and post-natal services. This study demonstrated improvements in access to and quality of basic services for mothers and neonates through a step-by-step institutionalization of innovative limited obstetric ultrasound services in semi-urban health centers in agrarian regions of Ethiopia.

**Table 4. Pairwise comparisons of ANC1, ANC4, SBA and PNC service coverages over four years (July 2016 –June 2020).**

| Sample 1-Sample 2 | Test statistic | Std. error | Std. test statistic | Sig. | Adj. sig.[a] |
|---|---|---|---|---|---|
| **ANC1** | | | | | |
| **2017–2018** | -8.167 | 8.979 | -0.909 | 0.363 | 1 |
| **2017–2020** | -29.317 | 8.979 | -3.265 | 0.001 | **0.007** |
| **2017–2019** | -30.05 | 8.979 | -3.347 | 0.001 | **0.005** |
| **2018–2020** | -21.15 | 8.979 | -2.355 | 0.019 | 0.111 |
| **2018–2019** | -21.883 | 8.979 | -2.437 | 0.015 | 0.089 |
| **2020–2019** | 0.733 | 8.979 | 0.082 | 0.935 | 1 |
| **ANC4** | | | | | |
| **2017–2018** | -9.367 | 8.98 | -1.043 | 0.297 | 1 |
| **2017–2020** | -24.5 | 8.98 | -2.728 | 0.006 | 0.038 |
| **2017–2019** | -32.867 | 8.98 | -3.66 | 0.001 | **0.002** |
| **2018–2020** | -15.133 | 8.98 | -1.685 | 0.092 | 0.552 |
| **2018–2019** | -23.5 | 8.98 | -2.617 | 0.009 | 0.053 |
| **2020–2019** | 8.367 | 8.98 | 0.932 | 0.351 | 1 |
| **SBA** | | | | | |
| **2017–2018** | -10.7 | 8.98 | -1.192 | 0.233 | 1 |
| **2017–2020** | -35.15 | 8.98 | -3.914 | 0.001 | **0.001** |
| **2017–2019** | -35.35 | 8.98 | -3.937 | 0.001 | **0.001** |
| **2018–2020** | -24.45 | 8.98 | -2.723 | 0.006 | 0.039 |
| **2018–2019** | -24.65 | 8.98 | -2.745 | 0.006 | 0.036 |
| **2020–2019** | -0.2 | 8.98 | -0.022 | 0.982 | 1 |
| **PNC** | | | | | |
| **2017–2018** | -5.8 | 8.981 | -0.646 | 0.518 | 1 |
| **2017–2020** | -27.583 | 8.981 | -3.071 | 0.002 | 0.013 |
| **2017–2019** | -30.95 | 8.981 | -3.446 | 0.001 | **0.003** |
| **2018–2020** | -21.783 | 8.981 | -2.425 | 0.015 | 0.092 |
| **2018–2019** | -25.15 | 8.981 | -2.8 | 0.005 | 0.031 |
| **2020–2019** | -3.367 | 8.981 | -0.375 | 0.708 | 1 |

Each row tests the null hypothesis that the Sample 1 and Sample 2 distributions are the same.

Asymptotic significances (2-sided tests) are displayed. The significance level is .05.

a. significance values have been adjusted by the Bonferroni correction for multiple tests.

The results of this study reveal an increased and statistically significant difference in first and fourth ANC service utilization. Increased coverage of ANC with ultrasound scanning services at twelve weeks of pregnancy helps to identify high risk health conditions including congenital anomalies, ectopic pregnancies, and abortion [6, 25]. In addition, ANC4 helps to determine other life-threatening conditions in women and neonates which includes mal-presentations, multiple fetuses, abnormalities in sizes of fetuses for gestational age, abnormal placentation, and antepartum hemorrhage [6, 25]. Hence, providing comprehensive ANC services contributes to the reduction of preventable maternal and neonatal deaths.

In this study, ANC1 and ANC4 service utilization rates were increased by one-third each. These findings are in alignment with studies conducted in both agrarian and pastoral regions of Ethiopia [26, 35]. Similarly, a scoping review has shown that the introduction of point of care ultrasound services into routine ANC resulted in higher ANC attendance [36]. A study in Uganda showed that the rate of ANC attendance was higher where portable ultrasounds were advertised, and women can be motivated to attend ANC visits when offered the concrete

incentive of seeing their baby [22]. Another study in Uganda showed that the introduction of a low-cost antenatal ultrasound program at a healthcare clinic in rural Uganda was associated with increases in the monthly mean number of ANC visits and increases in the number of women receiving specific recommended ANC interventions [37]. A study in Tanzania has also shown that the introduction of routine ultrasound scanning during ANC visits significantly increased the percentage of women attending ANC clinics by four times or more [15]. In contrast however, the trends of ANC4 coverages showed reductions from 2019 to 2020. This might be a result of interruption of services due to stock-out of essential supplies and turnover of trained professionals. The institutionalization of portable ultrasound innovation service is an invaluable asset in semi-urban or rural health centers where most perinatal and antenatal care of pregnant women are administered. These mothers usually lack access to better services available in referral health facilities, which are usually located in big cities, and decline lifesaving services due to fear of associated costs like transport, meals, accommodation, and consultation fee of traditional ultrasound machines [31, 35].

Similarly, the institutionalization of limited obstetric ultrasound services at semi urban or rural health centers has increased utilization of SBA and PNC services at time of delivery and immediately after to forty five days. Increased SBA and PNC services improve diagnoses and management of post-partum hemorrhage and very low birthweight cases ultimately curbing deaths of mothers and neonates, respectively [25]. These findings are also in alignment with studies conducted in other settings. The above-mentioned study in rural Uganda showed that following the introduction of ultrasound services, significant increases were seen in the number of mean monthly deliveries [37]. The study in Tanzania has also shown that the introduction of a simplified ultrasound scanning technology at the lowest levels of care has an effect of motivating women to select health facility deliveries [15]. A study conducted in Ghana showed that the use of a portable ultrasound scan during ANC increased the number of health facility deliveries [38]. Furthermore, a study in rural Eastern China has shown a statistically significant association between antenatal ultrasound scans and the uptake of cesarean section procedures [39].

The findings of this study demonstrate that improving the quality of services from pregnancy through to the perinatal period increased utilization of maternal and neonatal health services. The task shifting of ultrasound scanning services from senior ultra-sonographers to naïve mid-level health professionals improves access to quality maternal and neonatal health services in rural set-ups in Ethiopia [39, 40]. In addition, the information generated helps policy makers, program managers and healthcare workers to institutionalize and scale up limited obstetric ultrasound services in health centers and other emergency health service points within Ethiopia and other low-income countries.

## Limitations

One of the regions where Vscan access limited obstetric ultrasound services were introduced i.e., Tigray, was not included in the study as services were interrupted due to security reasons. Since this study employed a retrospective observational study design, it has a known limitation in regards to claiming causalities. In addition, all possible confounding factors like maternal and newborn interventions and socio-cultural and community factors could not be captured as would have been the case in prospective, randomized, and controlled trial studies.

## Conclusions and recommendations

Ethiopia is one of the countries in the world with low ANC, SBA, and PNC coverages. The introduction of limited obstetric ultrasound services by trained mid-level providers at health

centers was found to have improved the utilization of ANC, SBA, and PNC services. Hence, it is recommended that the institutionalization of limited obstetric ultrasound services by trained mid-level providers at health centers be scaled up. A further comparative study between facilities with and without ultrasound services to confirm causalities and to assess the effects on maternal and perinatal health outcomes is also recommended.

## Supporting information

**S1 File. Data file.**
(XLSX)

**S2 File. Statistical assumptions.**
(DOCX)

## Acknowledgments

We are indebted to all data managers, data collectors, supervisors, and health facility heads for their meticulous work in this research activity.

## Author Contributions

**Conceptualization:** Hailemariam Segni Abawollo, Mesele Damte Argaw, Zergu Taffesse Tsegaye, Ismael Ali Beshir, Binyam Fekadu Desta.

**Data curation:** Asfaw Adugna Guteta, Atrie Fekadu Heyi, Birhan Tenaw Mengesha, Tsega Teferi Mamo, Zenawork Kassa Gebremedhin, Heran Demissie Damte.

**Formal analysis:** Hailemariam Segni Abawollo, Mesele Damte Argaw, Zergu Taffesse Tsegaye.

**Funding acquisition:** Mengistu Asnake Kibret, Binyam Fekadu Desta.

**Investigation:** Hailemariam Segni Abawollo, Mesele Damte Argaw, Zergu Taffesse Tsegaye, Ismael Ali Beshir, Asfaw Adugna Guteta, Atrie Fekadu Heyi, Birhan Tenaw Mengesha, Tsega Teferi Mamo, Zenawork Kassa Gebremedhin, Heran Demissie Damte, Binyam Fekadu Desta.

**Methodology:** Hailemariam Segni Abawollo, Mesele Damte Argaw, Zergu Taffesse Tsegaye, Ismael Ali Beshir, Asfaw Adugna Guteta, Atrie Fekadu Heyi, Birhan Tenaw Mengesha, Tsega Teferi Mamo, Zenawork Kassa Gebremedhin, Heran Demissie Damte, Mengistu Asnake Kibret, Binyam Fekadu Desta.

**Project administration:** Mengistu Asnake Kibret, Binyam Fekadu Desta.

**Resources:** Mesele Damte Argaw, Mengistu Asnake Kibret, Binyam Fekadu Desta.

**Software:** Hailemariam Segni Abawollo, Mesele Damte Argaw.

**Supervision:** Asfaw Adugna Guteta, Atrie Fekadu Heyi, Birhan Tenaw Mengesha, Tsega Teferi Mamo, Zenawork Kassa Gebremedhin, Heran Demissie Damte.

**Validation:** Mesele Damte Argaw, Zergu Taffesse Tsegaye, Ismael Ali Beshir, Atrie Fekadu Heyi, Birhan Tenaw Mengesha, Tsega Teferi Mamo, Zenawork Kassa Gebremedhin, Heran Demissie Damte, Mengistu Asnake Kibret, Binyam Fekadu Desta.

**Visualization:** Mesele Damte Argaw.

**Writing – original draft:** Hailemariam Segni Abawollo, Mesele Damte Argaw.

**Writing – review & editing:** Hailemariam Segni Abawollo, Mesele Damte Argaw, Zergu Taffesse Tsegaye, Ismael Ali Beshir, Asfaw Adugna Guteta, Atrie Fekadu Heyi, Birhan Tenaw Mengesha, Tsega Teferi Mamo, Zenawork Kassa Gebremedhin, Heran Demissie Damte, Mengistu Asnake Kibret, Binyam Fekadu Desta.

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
