## [Decision Letter · Decision Letter 0]

24 Aug 2022

PONE-D-22-17217

Institutionalization of limited obstetric ultrasound program leads to increased antenatal, skilled delivery and postnatal service utilization in 3 regions of Ethiopia: a pre-post study

PLOS ONE

Dear Dr. Argaw,

Thank you for submitting your manuscript to PLOS ONE. After careful consideration, we feel that it has merit but does not fully meet PLOS ONE’s publication criteria as it currently stands. Therefore, we invite you to submit a revised version of the manuscript that addresses the points raised during the review process.

We look forward to receiving your revised manuscript.

Kind regards,

Godwin Otuodichinma Akaba, MBBS,MSc,MPH,FWACS

Academic Editor

PLOS ONE

Journal Requirements:

   "This study was fully funded by USAID as part of USAID Transform: Primary Health Care project, implemented under cooperative agreement number of AID-663-A-17-00002 in Ethiopia. The funding body had no role in the design, data collection, analysis, interpretation, and writing stages of the study’s undertaking."

   "The authors would like to acknowledge USAID for funding the study. We are indebted to data managers, data collectors, supervisors, and health facility heads."

 "This study was fully funded by USAID as part of USAID Transform: Primary Health Care project, implemented under cooperative agreement number of AID-663-A-17-00002 in Ethiopia. The funding body had no role in the design, data collection, analysis, interpretation, and writing stages of the study’s undertaking."

Additional Editor Comments:

This manuscript which reports the outcome of an intervention that deployed portable ultrasound scan services for use by midlevel providers at rural health facilities towards improving utilization MNCH services is quite relevant.

The manuscript has been fairly well written with inclusion relevant information needed for the understanding of the intervention deployed and how the outcomes were measured.

The following observations however needs to be addressed to improve the quality of the manuscript.

Background

Page 3, line 63: counties-Correct to countries

Page 4, line 92: compare-Please delete

Methods

Page 5, line 124: Please include the months during which the interventions were carried out. See also page 6, line133.

Page 8. line 176: were compared-Please delete

Results

Page 10, lines 215-223:

Comment: The characteristics of the study population should be described in the methodology section and not the result section. Please mention this as a subsection in materials and methods.

Maternal and neonatal services utilization

Comment: This section has mainly compared 2017 and 2020 without considering the trend in between these years. The authors should mention the decline in utilization when observed and explanations offered in the discussion section

Page 11, lines 223-224:

Comment: Please rephrase to: The coverage of first ANC showed an increasing trend from 43.6% in 2017 to 72,9% in 2020.

Table 2: Please include P-value at the relevant components of the table

Discussion

This section needs to be extensively revised as there was no robust discussion of the four maternal and newborn health services being studied. For example, ANC improved following the intervention apart from comparing with previous studies what is the implication of improved ANC attendance to maternal and newborn health. Is it possible that the benefit of satisfaction gained by the pregnant woman from the new technology encouraged her to deliver in the hospital under the supervision of a SBA attendant and subsequently attend follow up at PNC? Were the ultrasound scan services provided free to the patients?

The ultrasound scan could have enhanced diagnosis of impending complications and prompt referrals and interventions. How does this align with the overall goal of project? This is a health system strengthening intervention that has shifted a task to breach dearth of human resources for health in Ethiopia, what are the key lessons learnt? Can these lessons be considered in other relevant areas in service delivery in MNCH like in gynaecological emergencies for diagnosis of ectopic pregnancies and abortions etc?

Page 13, lines 259-263

Comment: These are repetitions from the introduction and not necessary in the discussion section.

Page 13, lines 263-267

Comment: Consider moving this towards the end of the discussion section as this relates to transforming research findings to policy and practice and usually should come after the discussion of findings from the study.

Limitation

Authors should also consider highlighting in the manuscript if there are other maternal and newborn health interventions by other agencies in these hospital during the study periods that may have affected the quality of care; as change in utilization of MNCH services can be shaped by other sociocultural and community factors which have not been mentioned in this manuscript.

On a general note, the reviewers have raised concerns in the following areas:

1.Staistical analysis and need to include the P-values in table 2

2.Description of the study population characteristics under the materials and methods section and not in the result section

2.Improvement in the discussion section

Reviewers' comments:

Reviewer's Responses to Questions

**Comments to the Author**

1. Is the manuscript technically sound, and do the data support the conclusions?

Reviewer #1: Partly

Reviewer #2: Partly

2. Has the statistical analysis been performed appropriately and rigorously? 

Reviewer #1: No

Reviewer #2: Yes

3. Have the authors made all data underlying the findings in their manuscript fully available?

Reviewer #1: No

Reviewer #2: Yes

4. Is the manuscript presented in an intelligible fashion and written in standard English?

Reviewer #1: Yes

Reviewer #2: Yes

5. Review Comments to the Author

Reviewer #1: REVIEW OF THE ARTICLE

INSTITUTIONALIZATION OF LIMITED ULTRASOUND PROGRAM LEADS TO INCREASED ANTENATAL, SKILLED DELIVERY AND POST NATAL SERVICE UTILIZATION IN THREE REGIONS OF ETHIOPIA: A PRE POST STUDY

Thank you for asking me to review the above article.

1. The authors have conducted research looking at interventions to reduce maternal mortality by increasing antenatal, skilled delivery and post natal services. This is a very laudable and ambitious plan. Also inculcating task shifting/sharing with mid-level providers as a part of the protocol is novel and forward thinking

2. The title is well articulated and self-explanatory.

3. The authors have chosen to use a structured abstract which is concise, and written in very plain language. All aspects of the study are included within the abstract

4. Line 92 the second ‘compare’ at the end should be expunged

Methodology#

1. The authors have described their study population but this should be better placed under methodology

2. The sampling technique is appropriate and clearly written and shows how the thirty health centers were chosen, the number of ultrasound machines deployed and how the midlevel staff were all trained.

3. The Authors have chosen to use an equivalent non parametric test to ANOVA for their data analyses . The Kruskall Wallis test is appropriate in that regard since more than 2 groups are being analysed.

4. The groups are segregated by years of service. The Kruskall Wallis test of ranks therefore seeks to define whether there is any difference in the means of the ranked data among the groups.

Each group has 4 categories viz ANC1, ANC4, skilled delivery and post natal care. The presentation of the data is a bit confusing for me, as I would have expected each outcome(ANC1, ANC4, skilled delivery and post natal care )for the 4 years to have been subjected to the KW test, and any significant difference, or lack thereof would be clearly stated.

To define which of the years had a more significant difference from the others , a post hoc test would then be applied to showcase this difference.

5. The data was further disaggregated into pre and post introduction of ultrasound . The years 2017 and 2018 represented the years prior to the introduction of the intervention while 2019 and 2020 represented the years after the introduction of said intervention. (lines 189 to 193). At this point a null and alternative hypotheses would have been nice so as to define exactly the desired outcomes of the study however, Having divided the data into 2 independent groups any statistical difference would be explored using the independent sample T test. The authors had correctly stated that this was not a normally distributed sample therefore the corresponding non parametric test (Mann Whitney U test) might have served better than the Kruskall Wallis test in this instance. The data had already been ranked and as such the rank sum would be a fairly straightforward calculation, with subsequent derivation of the P value using the software specified .

6. I would advise the authors should include a table showcasing the rank sum and mean rank per facility. Also p values should be added where necessary in the various tables

7. The discussion is well articulated and the direction of the argument of the authors is clear and well understood

Summary

This is a very well thought out study with interesting results and outcomes. The potential to positively influence health service delivery is massive with the ultimate goal of reducing maternal mortality.

There are however a few challenges in my opinion of the statistical methods used and presentation of data. The study will be much more robust if these suggestions are implemented

Thank you for the opportunity afforded me to review this article

Reviewer #2: This is a good manuscript describing the obstetrics outcomes improvement interventions. The manuscript is well written with good layout. However, the discussion section failed to adequately elaborate on the findings of the study. It is suggested that the authors take time to relate the findings with findings from similar studies and relate with the objectives of the process that is being reviewed.

My other comments are as highlighted with information for the attentions of the authors in the added sticky notes.

6. PLOS authors have the option to publish the peer review history of their article (what does this mean?). If published, this will include your full peer review and any attached files.

Reviewer #1: No

Reviewer #2: No

---

## [Author Response · Author response to Decision Letter 0]

4 Oct 2022

Reviewer comments and authors responses 

Journal requirements 

Response: Thank you so much for sharing the guides and links. Our manuscript has been corrected accordingly

2. Thank you for stating the following financial disclosure: "This study was fully funded by USAID as part of USAID Transform: Primary Health Care project, implemented under cooperative agreement number of AID-663-A-17-00002 in Ethiopia. The funding body had no role in the design, data collection, analysis, interpretation, and writing stages of the study’s undertaking."

Response: Thank you so much for the comments and suggestions. The Financial disclosure statement is revised as presented below. 

This study was fully funded by USAID as part of USAID Transform: Primary Health Care project, implemented by Pathfinder International and JSI Research & Training Institute Inc under cooperative agreement number of AID-663-A-17-00002 in Ethiopia. The project purchased Vscan limited obstetric ultrasound machines and distributed it with seed supplies to 100 health centers. The cost of basic ultrasound training was also covered by the project. In addition, the project covers the cost of data collectors. The funder provided support in the form of salaries for authors HAS, MDA, ZTT, IAB, AAG, AFH, BTM, TTM, ZKG, HDD, MAK, and BFD. However, the funding body had no role in the study design, data collection, analysis, decision to publish, or preparation of the manuscript.

"The authors would like to acknowledge USAID for funding the study. We are indebted to data managers, data collectors, supervisors, and health facility heads." We note that you have provided funding information that is not currently declared in your Funding Statement. However, funding information should not appear in the Acknowledgments section or other areas of your manuscript. 

Response: Thank you so much for the comments and suggestions. The funding information is deleted form acknowledgements and rephrased as follow. 

Acknowledgments

We are indebted to all data managers, data collectors, supervisors, and health facility heads for their meticulous work in this research activity.

4. In your Data Availability statement, you have not specified where the minimal data set underlying the results described in your manuscript can be found. PLOS defines a study's minimal data set as the underlying data used to reach the conclusions drawn in the manuscript and any additional data required to replicate the reported study findings in their entirety. All PLOS journals require that the minimal data set be made fully available. For more information about our data policy, please see http://journals.plos.org/plosone/s/data-availability

Response: Thank you so much for the comments. Please update the data availability statement as follow. 

All relevant data are within the manuscript and supporting information files. 

Response: Thank you so much for the comments. Please see the response for Comment # 4 above. 

Response: Thank you so much for the comments. All supporting information are presented at the end of the manuscript and in-test citations. 

Supplementary information

S1 Fig 1.: Bar chart showing first ANC, fourth ANC, SBA, and PNC service utilization coverages prior to, and post the introduction of ultrasound services at health centers, Ethiopia, 2017-2020.

S1 Table 1.: Characteristics of the study area and population, USAID Transform: Primary health Care project intervention sites, Ethiopia, 2017- 2020.

S2 Table 2.: Number of monthly ANC, SBA, and PNC services utilization in 30 health centers pre- and post-ultrasound introduction in Ethiopia, 2017-2020.

S3 Table 3.: Mean rank scores of ANC1, ANC4, SBA, PNC services and test of difference in its distribution over four years (2017- 2020)

S4 Table 4.: Pairwise comparisons of ANC1 ANC4 Delivery and PNC service coverages over four years (2017 – 2020)

S1 File 1.: Data file

2 File 2.: Statistical assumptions

Summary of the research and overall impression 

The reviewers acknowledged the importance of this novel research work and appreciate the effort made to understand the intervention deployed and measured outcomes. The study is very laudable and inculcate the importance of task shifting/sharing to improve quality of services in different point of care. 

Discussion on specific area of improvement 

Introduction 

1. Line 143: Objective - the second objective is to “develop malaria care service modalities……” but nothing was developed from the paper, rather than this paper provide input for development of 

The wording of the aim of this formative assessment now has been updated. The last paragraph of the introduction, lines 109-112 now read as:

“We conducted a formative assessment to collect baseline data on seasonal migrant workers access to malaria prevention and control interventions. The data generated will provide empirical evidence for the development of malaria care service modalities and elimination strategies in seasonal migrant/mobile workers in Ethiopia.”

Background

2. Page 3, line 63: counties-Correct to countries

Thank you so much for the comment. On page 3, line 63, the word “countries” is corrected.

3. Page 4, line 92: compare-Please delete

Thank you so much for the comment. On page 4, line 92, the second “compare” is deleted.

Methods

4. Page 5, line 124: Please include the months during which the interventions were carried out. See also page 6, line133.

Thank you so much for the comment. On page 5, line 124, and page 6, line 133 the months of interventions were included. 

5. Page 8. line 176: were compared-Please delete

Thank you so much for the comment. On page 8, line 186, the word compare is deleted. 

Results

6. Page 10, lines 215-223:

Comment: The characteristics of the study population should be described in the methodology section and not the result section. Please mention this as a subsection in materials and methods.

Maternal and neonatal services utilization

Thank you so much for the comment. The study population is moved to methodology section. Please see the changes on page 78, lines 156 – 165. 

7. Comment: This section has mainly compared 2017 and 2020 without considering the trend in between these years. The authors should mention the decline in utilization when observed and explanations offered in the discussion section

Thank you so much for the comment. On page 12 lines 256- 257, the mean fourth ANC coverage showed a reduction from 76.68% to 68.32%. This might be due to interruption of services due to stockout of essential supplies and turnover of trained professionals. On page 16, lines 328 – 330, the discussion section included the following information.

However, the trends of ANC4 coverages showed reduction from 2019 to 2020.This might be happed due to interruption of services due to stockout of essential supplies and turnover of trained professionals.

Page 11, lines 223-224:

8. Comment: Please rephrase to: The coverage of first ANC showed an increasing trend from 43.6% in 2017 to 72,9% in 2020

Thank you so much for the comment. The information of maternal and neonatal health services is rephrased. Please see changes on page11 lines 241- 247.

9. Table 2: Please include P-value at the relevant components of the table

Thank you so much for the comment. Table 3 presents the detail information of KW- ANOVA test results. Please see changes on page13 lines 272- 274.

Discussion

10. This section needs to be extensively revised as there was no robust discussion of the four maternal and newborn health services being studied. For example, ANC improved following the intervention apart from comparing with previous studies what is the implication of improved ANC attendance to maternal and newborn health. Is it possible that the benefit of satisfaction gained by the pregnant woman from the new technology encouraged her to deliver in the hospital under the supervision of a SBA attendant and subsequently attend follow up at PNC? Were the ultrasound scan services provided free to the patients?

The ultrasound scan could have enhanced diagnosis of impending complications and prompt referrals and interventions. How does this align with the overall goal of project? This is a health system strengthening intervention that has shifted a task to breach dearth of human resources for health in Ethiopia, what are the key lessons learnt? Can these lessons be considered in other relevant areas in service delivery in MNCH like in gynecological emergencies for diagnosis of ectopic pregnancies and abortions etc?

Thank you so much for the comment. The discussion section revised, and detail information are included. The following changes are made. Please see changed on page13 lines 294 – 298, 308 – 310, 313- 317, 328- 330.

11. Page 13, lines 259-263

Comment: These are repetitions from the introduction and not necessary in the discussion section.

Thank you so much for the comment. The first paragraph of the discussion is removed.

12. Page 13, lines 263-267

Comment: Consider moving this towards the end of the discussion section as this relates to transforming research findings to policy and practice and usually should come after the discussion of findings from the study.

Thank you so much for the comment. The information is moved to the end of the discussion section. On Page 16, lines 325- 330.

Limitation

13. Authors should also consider highlighting in the manuscript if there are other maternal and newborn health interventions by other agencies in these hospital during the study periods that may have affected the quality of care; as change in utilization of MNCH services can be shaped by other sociocultural and community factors which have not been mentioned in this manuscript.

Thank you so much for the comment. The limitation of the study is rephrased. See changes on page 16, lines 330 -336.

14. On a general note, the reviewers have raised concerns in the following areas:

• Statistical analysis and need to include the P-values in table 2

• Description of the study population characteristics under the materials and methods section and not in the result section

• Improvement in the discussion section

Thank you so much for the comment. All the three comments are properly addressed.

Reviewer's Responses to Questions

Comments to the Author

1. Is the manuscript technically sound, and do the data support the conclusions?

Reviewer #1: Partly

Reviewer #2: Partly

Thank you so much for the comment. The revised manuscript will satisfy reviewers in the current revised form.

2. Has the statistical analysis been performed appropriately and rigorously?

Reviewer #1: No

Reviewer #2: Yes

Thank you so much for the comment. The statistical test includes the comments of reviewers. We hope, the reviewers will be satisfied in the revised form.

3. Have the authors made all data underlying the findings in their manuscript fully available?

Reviewer #1: No

Reviewer #2: Yes

Thank you so much for the comment. All relevant data are within the manuscript and supporting information files. 

4. Is the manuscript presented in an intelligible fashion and written in standard English?

Reviewer #1: Yes

Reviewer #2: Yes

Thank you so much for the encouraging comments.

5. Review Comments to the Author

Reviewer #1: REVIEW OF THE ARTICLE

Thank you for asking me to review the above article.

15. The authors have conducted research looking at interventions to reduce maternal mortality by increasing antenatal, skilled delivery and post-natal services. This is a very laudable and ambitious plan. Also inculcating task shifting/sharing with mid-level providers as a part of the protocol is novel and forward thinking

Thank you so much for the encouraging comments.

16. The title is well articulated and self-explanatory.

Thank you so much for the encouraging comments.

17. The authors have chosen to use a structured abstract which is concise and written in very plain language. All aspects of the study are included within the abstract

Thank you so much for the encouraging comments.

18. Line 92 the second ‘compare’ at the end should be expunged

Thank you so much for the comments. The second compare is deleted now.

Methodology#

19. The authors have described their study population, but this should be better placed under methodology

Thank you so much for the comments. The description of the study population is moved to methodology section. Please see the changes on page 78, lines 156 – 165. 

20. The sampling technique is appropriate and clearly written and shows how the thirty health centers were chosen, the number of ultrasound machines deployed and how the midlevel staff were all trained.

Thank you so much for the encouraging comments.

21. The Authors have chosen to use an equivalent nonparametric test to ANOVA for their data analyses. The Kruskal Wallis test is appropriate in that regard since more than 2 groups are being analyzed. The groups are segregated by years of service. The Kruskal Wallis test of ranks therefore seeks to define whether there is any difference in the means of the ranked data among the groups. Each group has 4 categories viz ANC1, ANC4, skilled delivery and post-natal care. The presentation of the data is a bit confusing for me, as I would have expected each outcome(ANC1, ANC4, skilled delivery and post-natal care )for the 4 years to have been subjected to the KW test, and any significant difference, or lack thereof would be clearly stated.

To define which of the years had a more significant difference from the others , a post hoc test would then be applied to showcase this difference.

Thank you so much for the comments. Two tables on Kruskal Wallis H test and post hoc tests are included in the current form. 

22. The data was further disaggregated into pre and post introduction of ultrasound. The years 2017 and 2018 represented the years prior to the introduction of the intervention while 2019 and 2020 represented the years after the introduction of said intervention. (Lines 189 to 193). At this point a null and alternative hypothesis would have been nice so as to define exactly the desired outcomes of the study however, having divided the data into 2 independent groups any statistical difference would be explored using the independent sample T test. The authors had correctly stated that this was not a normally distributed sample therefore the corresponding nonparametric test (Mann Whitney U test) might have served better than the Kruskal Wallis test in this instance. The data had already been ranked and as such the rank sum would be a fairly straightforward calculation, with subsequent derivation of the P value using the software specified.

Thank you so much for the comments. Since the authors want to determine distribution of maternal and neonatal health service coverages over four years period, we prefer to work on Kruskal Wallis H test and post hoc tests than dividing the data into two.

23. I would advise the authors should include a table showcasing the rank sum and mean rank per facility. Also, p values should be added where necessary in the various tables

Thank you so much for the comments. Table 3 and 4 have presented the mean rank scores and associated p-values.

24. The discussion is well articulated and the direction of the argument of the authors is clear and well understood

Thank you so much for the encouraging comments.

Reviewer #2: 

25. This is a good manuscript describing the obstetrics outcomes improvement interventions. The manuscript is well written with good layout. However, the discussion section failed to adequately elaborate on the findings of the study. It is suggested that the authors take time to relate the findings with findings from similar studies and relate with the objectives of the process that is being reviewed.

Thank you so much for the encouraging comments.

26. My other comments are as highlighted with information for the attentions of the authors in the added sticky notes.

Which cadre of healthcare workers constitute this mid-level health professionals?

Thank you so much for the comments. The mid-level health professionals are BSc nurses and midwives.

When in 2017 did the implementation of the intervention start and when in 2020 did the intervention end?

Thank you so much for the comments. The study period was July 2016 through June 2020. 

What is GE?

Thank you so much for the comments. GE is presented in its expanded form General Electric.

Recast this statement as this read like 100 ultrasound machines were provided to each of the health center!

Thank you so much for the comments. The sentence id rephrased as follow: 

”Vscan ultrasound machine with seed supplies were provided to each targeted 100 health centers,”

What accounted for this 3.4 times drop between first and second ultrasound visits?

Thank you so much for the comments. The US scanning services is recommended to be conducted on 13 weeks of gestation to diagnose gross anomalies. And the second scanning services is after 24 weeks to rule out high risk conditions. The significant difference on US 1, US2 and US3 scanning services were in line with the reduction of subsequent ANC service utilization.

Delete the phrase “were determined” 

Thank you so much for the comments and the phrase is deleted.

The discussion section needs to be greatly improved on to bring out the findings of this study and compare with similar studies!

Thank you so much for the comments, The discussion is revised and improved in the current form.

---

## [Decision Letter · Decision Letter 1]

17 Jan 2023

PONE-D-22-17217R1Institutionalization of limited obstetric ultrasound program leads to increased antenatal, skilled delivery and postnatal service utilization in 3 regions of Ethiopia: a pre-post studyPLOS ONE

Dear Dr. Argaw,

Thank you for submitting your manuscript to PLOS ONE. After careful consideration, we feel that it has merit but does not fully meet PLOS ONE’s publication criteria as it currently stands. Therefore, we invite you to submit a revised version of the manuscript that addresses the points raised during the review process.

We look forward to receiving your revised manuscript.

Kind regards,

Ikechukwu Innocent Mbachu

Academic Editor

PLOS ONE

Journal Requirements:

Additional Editor Comments (if provided):

Thank you for the detailed revision of the manuscript.

The manuscript will be considered for publication after your response to the minor revision raised by the reviewers

Reviewers' comments:

Reviewer's Responses to Questions

**Comments to the Author**

1. If the authors have adequately addressed your comments raised in a previous round of review and you feel that this manuscript is now acceptable for publication, you may indicate that here to bypass the “Comments to the Author” section, enter your conflict of interest statement in the “Confidential to Editor” section, and submit your "Accept" recommendation.

Reviewer #1: All comments have been addressed

Reviewer #3: All comments have been addressed

2. Is the manuscript technically sound, and do the data support the conclusions?

Reviewer #1: Yes

Reviewer #3: Yes

3. Has the statistical analysis been performed appropriately and rigorously? 

Reviewer #1: Yes

Reviewer #3: Yes

4. Have the authors made all data underlying the findings in their manuscript fully available?

Reviewer #1: Yes

Reviewer #3: No

5. Is the manuscript presented in an intelligible fashion and written in standard English?

Reviewer #1: Yes

Reviewer #3: No

6. Review Comments to the Author

Reviewer #1: COMMENTS ON THE ARTICLE ‘INSTITUTIONALISATION OF LIMITED OBSTETRIC ULTRASOUND PROGRAM LEADS TO INCREASED ANTENATAL SKILLED DELIVERY AND POSTNATAL SERVICE UTILASITION IN 3 REGIONS OF ETHIOPA: A PRE POST STUDY’

THANK YOU FOR ASKING ME TO REVIEW THIS ARTICLE

1. THIS ARTICLE IS FRESH FORWARD THINKING AND RELEVANT AS A STRATEGY TO REDUCE MATERNAL MORTALITY

2. PLEASE CHANGE ‘HEALTH SERVICES ‘IN LINE 64 TO ‘HEALTH INTERVENTIONS’

3. WAS THERE SIGNIFICANT SYNCHRONY WITH THE ULTRASOUND DIAGNOSES OF THE TRAINED MID LEVEL STAFF AND THE REFERRAL HOSPITALS TO MERIT CONTINUATION OF THE PROGRAM? (LINE 51)

4. THE STATISTICAL TESTS ARE APPROPRIATELY DEPLOYED

5. FROM LINE 235 TO 254 THE MEAN RANK SUMS OF THE VARIOUS INDICES ARE PRESENTED. ONLY YEARS 2017 AND 2020 ARE REFERENCED FOR ANC1, SBA AND PNC. ANC4 HAS YAER 2019 ADDED TO IT. ARE THE OTHER YEARS NOT PART OF THR ASSSESMENT TO CHECK FOR SIGNIFICANCE?

6. THE DISCUSSION NEEDS A BIT MORE DEPTH TO ENRICH THE ARTICLE.

THANK YOU

Reviewer #3: This study is a pre and post intervention observational study was conducted to investigate

maternal and neonatal health service utilization rates before and after institutionalizing

Vscan limited obstetric ultrasound services, between July 2016 and June 2020.

ABSTRACT

This is well written , unfortunately, the objectives of the study was missing in the abstract.

INTRODUCTION

The authors should beef up the introduction part of the study.

Methods

It is ok.

RESULTS

This is adequate

DISCUSSION

Fairly writen

7. PLOS authors have the option to publish the peer review history of their article (what does this mean?). If published, this will include your full peer review and any attached files.

Reviewer #1: No

Reviewer #3: **Yes: **George Eleje

---

## [Author Response · Author response to Decision Letter 1]

26 Jan 2023

Journal Requirements:

Comment accepted and the correct funding information is included in the cover latter. 

Response: Thank you so much for your critical comments. We checked all references are cited as per the authors guidelines and now to our knowledge no retracted articles are cited in the manuscript. We hope our improvement satisfies you.

Additional Editor Comments (if provided):

Thank you for the detailed revision of the manuscript.

The manuscript will be considered for publication after your response to the minor revision raised by the reviewers

Thank you so much for your acknowledgements on our effort. 

Reviewers' comments:

Comments to the Author

1. If the authors have adequately addressed your comments raised in a previous round of review and you feel that this manuscript is now acceptable for publication, you may indicate that here to bypass the “Comments to the Author” section, enter your conflict-of-interest statement in the “Confidential to Editor” section, and submit your "Accept" recommendation.

Reviewer #1: All comments have been addressed

Reviewer #3: All comments have been addressed

Thank you so much for your constructive feedback. 

2. Is the manuscript technically sound, and do the data support the conclusions?

Reviewer #1: Yes

Reviewer #3: Yes

Thank you so much for your constructive feedback. 

3. Has the statistical analysis been performed appropriately and rigorously?

Reviewer #1: Yes

Reviewer #3: Yes

Thank you so much for your constructive feedback. 

4. Have the authors made all data underlying the findings in their manuscript fully available?

Reviewer #1: Yes

Reviewer #3: No

Thank you so much for your comment. We already attached all the data analyzed to write this manuscript as a supplementary file (S1 File 1). 

5. Is the manuscript presented in an intelligible fashion and written in standard English?

Reviewer #1: Yes

Reviewer #3: No

Thank you so much for your comment. The manuscript is edited by a native English speaker. I hope now, it meets the journal requirement. 

6. Review Comments to the Author

Reviewer #1: COMMENTS ON THE ARTICLE ‘INSTITUTIONALISATION OF LIMITED OBSTETRIC ULTRASOUND PROGRAM LEADS TO INCREASED ANTENATAL SKILLED DELIVERY AND POSTNATAL SERVICE UTILASITION IN 3 REGIONS OF ETHIOPA: A PRE POST STUDY’

THANK YOU FOR ASKING ME TO REVIEW THIS ARTICLE

1. THIS ARTICLE IS FRESH FORWARD THINKING AND RELEVANT AS A STRATEGY TO REDUCE MATERNAL MORTALITY

Response: Thank you so much for your recognition. 

2. PLEASE CHANGE ‘HEALTH SERVICES ‘IN LINE 64 TO ‘HEALTH INTERVENTIONS’

Response: Thank you so much for your comment. On page line , the word ‘service’ is replaced with ‘intervention’. 

3. WAS THERE SIGNIFICANT SYNCHRONY WITH THE ULTRASOUND DIAGNOSES OF THE TRAINED MID LEVEL STAFF AND THE REFERRAL HOSPITALS TO MERIT CONTINUATION OF THE PROGRAM? (LINE 51)

Response: Thank you so much for your comment This is a retrospective study, it is stated in its statement of limitation. 

4. THE STATISTICAL TESTS ARE APPROPRIATELY DEPLOYED

Response: Thank you so much for your recognition. 

5. FROM LINE 235 TO 254 THE MEAN RANK SUMS OF THE VARIOUS INDICES ARE PRESENTED. ONLY YEARS 2017 AND 2020 ARE REFERENCED FOR ANC1, SBA AND PNC. ANC4 HAS YAER 2019 ADDED TO IT. ARE THE OTHER YEARS NOT PART OF THR ASSSESMENT TO CHECK FOR SIGNIFICANCE?

Response: Thank you so much for your comment. Now on page 11, lines 252-264, paragraph is rephrased to reflect results over four years.

The mean rank of first ANC visits was 43.62, 52.78, 73.67, and 72.93 for the years 2017, 2018, 2019, and 2020, respectively. The introduction of Vscan limited obstetric ultrasound service shows a statistically significant difference on first ANC visits over four years at KW-ANOVA H (3) = 17.09, P=0.001. Similarly, the mean rank of fourth ANC was 43.82, 53.18, 76.68, and 68.32 in 2017, 2018, 2019 and 2020, respectively. 

The mean rank SBA scores were 40.20, 50.90, 75.35, and 75.55in the year 2017, 2018, 2019 and 2020, respectively. The observed increased mean rank SBA scores show a statistically significant positive difference using KW-ANOVA H (3) = 23.6, P<0.001. While the mean rank of PNC service scores were 44.42, 50.22, 72.00, and 75.37 in the years 2017, 2018, 2019 and 2020, respectively. 

6. THE DISCUSSION NEEDS A BIT MORE DEPTH TO ENRICH THE ARTICLE.

Response: Thank you so much for your comment. The discussion is revised. We hope, the current form will satisfy the reviewers. See changes on page 12-14, lines 282- 288; 290-298; 309-319. 

This study has shown that the institutionalization of limited obstetric ultrasound services by trained mid-level providers at semi urban health centers significantly improved the utilization of prenatal, intrapartum, and post-natal services. This study demonstrated improvements in access to and quality of basic services for mothers and neonates through a step-by-step institutionalization of innovative limited obstetric ultrasound services in semi-urban health centers in agrarian regions of Ethiopia. 

The results of this study reveal an increased and statistically significant difference in first and fourth ANC service utilization. Increased coverage of ANC with ultrasound scanning services at twelve weeks of pregnancy helps to identify high risk health conditions including congenital anomalies, ectopic pregnancies, and abortion [6, 25]. In addition, ANC4 helps to determine other life-threatening conditions in women and neonates which includes mal-presentations, multiple fetuses, abnormalities in sizes of fetuses for gestational age, abnormal placentation, and antepartum hemorrhage [6, 25]. Hence, providing comprehensive ANC services contributes to the reduction of preventable maternal and neonatal deaths. 

In this study, ANC1 and ANC4 service utilization rates were increased by one-third each. These findings are in alignment with studies conducted in both agrarian and pastoral regions of Ethiopia [26, 35].

 The institutionalization of portable ultrasound innovation service is an invaluable asset in semi-urban or rural health centers where most perinatal and antenatal care of pregnant women are administered. These mothers usually lack access to better services available in referral health facilities, which are usually located in big cities, and decline lifesaving services due to fear of associated costs like transport, meals, accommodation, and consultation fee of traditional ultrasound machines [31, 35]. 

Similarly, the institutionalization of limited obstetric ultrasound services at semi urban or rural health centers has increased utilization of SBA and PNC services at time of delivery and immediately after to forty five days. Increased SBA and PNC services improve diagnoses and management of post-partum hemorrhage and very low birthweight cases ultimately curbing deaths of mothers and neonates, respectively [25].

Reviewer #3: This study is a pre and post intervention observational study was conducted to investigate maternal and neonatal health service utilization rates before and after institutionalizing Vscan limited obstetric ultrasound services, between July 2016 and June 2020.

ABSTRACT

This is well written, unfortunately, the objectives of the study was missing in the abstract.

Response: Thank you so much for your comment. Now, on page 2, lines 32- 35, the objective of the study is included in the abstract.

The aims of this study were to compare ANC1, ANC4, SBA, and PNC service utilization before and after institutionalizing Vscan limited obstetric ultrasounds at semi-urban health centers in Ethiopia.

INTRODUCTION

The authors should beef up the introduction part of the study.

Response: Thank you so much for your comment.

Methods

It is ok.

Response: Thank you so much for your recognition. 

RESULTS

This is adequate

Response: Thank you so much for your recognition. 

DISCUSSION

Fairly written

Response: Thank you so much for your recognition. 
---

## [Editor Report · Decision Letter 2]

30 Jan 2023

Institutionalization of limited obstetric ultrasound leading to increased antenatal, skilled delivery, and postnatal service utilization in three regions of Ethiopia: a pre-post study

PONE-D-22-17217R2

Dear Dr. Argaw,

We’re pleased to inform you that your manuscript has been judged scientifically suitable for publication and will be formally accepted for publication once it meets all outstanding technical requirements.

Kind regards,

Ikechukwu Innocent Mbachu

Academic Editor

PLOS ONE

Additional Editor Comments (optional):

The corrections have been noted.

The article is now acceptable for publication pending fulfillment of other journal's requirements.
---

## [Editor Report · Acceptance letter]

2 Feb 2023

PONE-D-22-17217R2 

Institutionalization of limited obstetric ultrasound leading to increased antenatal, skilled delivery, and postnatal service utilization in three regions of Ethiopia: a pre-post study 

Dear Dr. Argaw:

I'm pleased to inform you that your manuscript has been deemed suitable for publication in PLOS ONE. Congratulations! Your manuscript is now with our production department. 

Kind regards, 

on behalf of

Dr. Ikechukwu Innocent Mbachu 

Academic Editor

PLOS ONE